# Effects of Disturbance on Understory Vegetation across Slovenian Forest Ecosystems

**Lado Kutnar** [1,*]**, Thomas A. Nagel** [2] **and Janez Kermavnar** [1] 

1 Slovenian Forestry Institute, 1000 Ljubljana, Slovenia; janez.kermavnar@gozdis.si
2 Biotechnical Faculty, University of Ljubljana, 1000 Ljubljana, Slovenia; tom.nagel@bf.uni-lj.si
* Correspondence: lado.kutnar@gozdis.si

**Abstract:** The herbaceous understory represents a key component of forest biodiversity across temperate forests of Europe. Here, we quantified changes in the diversity and composition of the forest understory layer in representative Slovenian forest ecosystems between 2004/05 and 2014/15. In total, 60 plots were placed across 10 different managed forest types, ranging from lowland deciduous and mid-altitude mesic mixed forests to mountain conifer forests. This network is part of an international network of sites launched within the ICP Forests Programme aimed to assess the condition of forests in Europe. To examine how disturbance influenced understory dynamics, we estimated the disturbance impacts considering both natural and/or anthropogenic disturbances that cause significant damage to trees and to ground-surface layers, including ground-vegetation layers and upper-soil layers. Species richness across 10 sites (gamma diversity) significantly decreased from 272 to 243 species during the study period, while mean species richness per site did not significantly change. The mean value of site level Shannon diversity indices and evenness significantly increased. The cover of most common plant species increased during the monitoring period. The mean value of disturbance estimates per site increased from 0.8% in 2004/05 (ranging from 0% to 2.5%) to 16.3% in 2014/15 (ranging from 5.0% to 38.8%), which corresponded to a reduction in total vegetation cover, including tree-layer cover. More disturbed sites showed larger temporal changes in species composition compared to less disturbed sites, suggesting that forest disturbances caused understory compositional shifts during the study period. Rather than observing an increase in plant diversity due to disturbance, our results suggest a short-term decrease in species number, likely driven by replacement of more specialized species with common species.

**Keywords:** vegetation dynamics; vascular-plant diversity; understory layer; disturbance; monitoring; temperate forests

## 1. Introduction

Biodiversity in forestlands has become one of the major concerns of forestry in the European Union, as well as globally [1]. As conservation of biodiversity has become one of the important goals of managing forests in recent decades [2], variables and indicators related to biodiversity need to be monitored in forests, particularly during the current period of rapid global change.

Most of the plant diversity in temperate forest ecosystems is found in the understory herb layer. This diverse flora represents a wide variety of growth forms and functional groups, which collectively have an important influence on a variety of forest processes, including nutrient cycling and competitive interactions with tree regeneration [3]. The herb layer also serves as an important source of food and habitat for wildlife and other forms of biodiversity [4].

A large body of research over the past decade demonstrates that the understory herb layer exhibits strong temporal dynamics [5]. These dynamics have typically been attributed to several different anthropogenic drivers, including changes in light conditions due to forest management [6], nitrogen deposition [7–9], climate change [10,11], and excessive deer browsing [12,13].

Natural and anthropogenic disturbances in forest ecosystems are also important drivers of ecosystem structure, function, and biodiversity [14–16]. Disturbances can cause abrupt changes in understory light and resource availability and can have a lasting legacy on long-term forest dynamics, both of which influence understory herb communities [17]. Damage from wind, insect outbreaks and wildfires have increased in Europe's forests throughout the twentieth century and are critical drivers of the composition, structure and functioning of forest ecosystems [18,19].

A proper understanding of these various drivers and the response of the understory community form a foundation for plant biodiversity conservation in forests. Such an understanding often requires direct long-term monitoring of vegetation dynamics [20]. In Europe, systematic forest monitoring started in the 1970s as a response to questions related to the supposed forest decline caused by air pollution. These concerns led to an international effort resulting in the United Nations Convention on Long-range Transboundary Air Pollution (CLRTAP, Geneva 1979), and to the International Co-operative Programme on the Assessment and Monitoring of Air Pollution Effects on Forests (ICP Forests), which was launched in 1985 [21,22]. As a part of ICP Forests, the Intensive Forest Monitoring Programme was initiated to obtain sufficiently detailed information on processes within forest ecosystems across a broad spatial scale. The main goal of the monitoring programme is to detect and investigate the extent of changes within forest ecosystems due to the input of acidifying and eutrophying compounds [23]. Within the monitoring protocol, understory vegetation and biodiversity are observed across European forests [24] since changes in the abundance and composition of understory plant communities serve as useful indicators of ongoing changes in site conditions [8,25–28], or other important factors, such as disturbance, browsing, or climate change.

In Slovenia, Intensive Monitoring sites (IM) were established across the main forest types in homogeneous closed canopy stands lacking evidence of recent disturbances. However, over the past decade these sites have experienced a number of widespread disturbances of varying intensity, providing an opportunity to examine the short-term response of the understory community. The main aims of our study were (1) to determine the magnitude and direction of the changes in forest vegetation in managed temperate forests in Slovenia over the 10-year observation period; and to (2) examine the impact of natural disturbances, including salvage logging, on plant diversity and composition.

## 2. Materials and Methods

### 2.1. Study Area

The study was conducted in 10 IM sites (Table 1, Figure 1), which were systematically selected in all phyto-geographic regions in Slovenia [29] to represent the mayor forest ecosystems and forest vegetation communities across the country [30–32]. All IM sites were established in 2004 in closed canopy forest stands without any significant signs of recent disturbance. They cover an elevational gradient from 160 to 1397 m a.s.l. *Fagus sylvatica* L. is dominant at four sites, *Picea abies* (L.) H. Karst. at two sites, *Quercus robur* L. and *Carpinus betulus* L. at two sites, and *Pinus sylvestris* L. and *Pinus nigra* J.F.Arnold at one site each.

**Table 1.** Main characteristics of Intensive Monitoring (IM) sites in Slovenia [33,34].

| Name (Abbreviation) of IM Site | Area & Phyto-geographic Region [28] | Latitude (X) | Longitude (Y) | Elevation (m) | Temperature (10-Years Mean) (deg. C) | Bedrock | Soil Unit | Dominant Tree Species | Number of Plots |
|---|---|---|---|---|---|---|---|---|---|
| 1. KRUCMANOVE KONTE (1-KK) | Pokljuka, Alpine region | 418,719 | 136,466 | 1397 | 4.4 | Moraine | Eutric Cambisols, Rendzic Leptosols | *Picea abies* | 4 |
| 2. FONDEK (2-FO) | Trnovski gozd, Dinaric region | 402,239 | 95,690 | 827 | 11.2 | Limestone | Rendzic Leptosols, Eutric Cambisols | *Fagus sylvatica* | 8 |
| 3. GROPAJSKI BORI (3-GB) | Sežana, Sub-Mediterranean region | 411,589 | 59,052 | 420 | 11.9 | Limestone | Chromic Cambisols | *Pinus nigra* | 4 |
| 4. BRDO (4-BR) | Kranj, Pre-Alpine region | 454,133 | 127,146 | 471 | 10.3 | Fluvioglacial gravels and sands | Dystric Cambisols | *Pinus sylvestris* | 8 |
| 5. BOROVEC (5-BO) | Kočevska Reka, Dinaric region | 484,737 | 43,605 | 705 | 8.7 | Limestone, Dolomite | Rendzic Leptosols, Eutric Cambisols | *Fagus sylvatica* | 8 |
| 6. KLADJE (6-KL) | Pohorje, Alpine region | 530,522 | 147,809 | 1304 | 4.9 | Dioritoid (Tonalite) | Dystric Cambisols | *Picea abies* | 4 |
| 8. LONTOVŽ (8-LO) | Kum, Pre-Dinaric region | 505,362 | 105,871 | 958 | 7.4 | Dolomite | Rendzic Leptosols, Eutric Cambisols | *Fagus sylvatica* | 8 |
| 9. GORICA (9-GO) | Loški potok, Dinaric region | 471,818 | 54,755 | 955 | 8.2 | Dolomite | Rendzic Leptosols, Eutric Cambisols | *Fagus sylvatica* | 4 |
| 10. KRAKOVSKI GOZD (10-KG) | Kostanjevica na Krki, Sub-Pannonian region | 532,688 | 82,059 | 160 | 9.8 | Pleistocene sediments | Gleysols | *Carpinus betulus (Quercus robur)* | 4 |
| 11. MURSKA ŠUMA (11-MŠ) | Lendava, Sub-Pannonian region | 616,509 | 151,426 | 170 | 10.5 | Alluvium | Gleysols, Fluvisol | *Quercus robur (Carpinus betulus)* | 8 |

Note: Site number 7 Vinska gora (Dobrna) was studied only in 2004 and was not included in the current study.

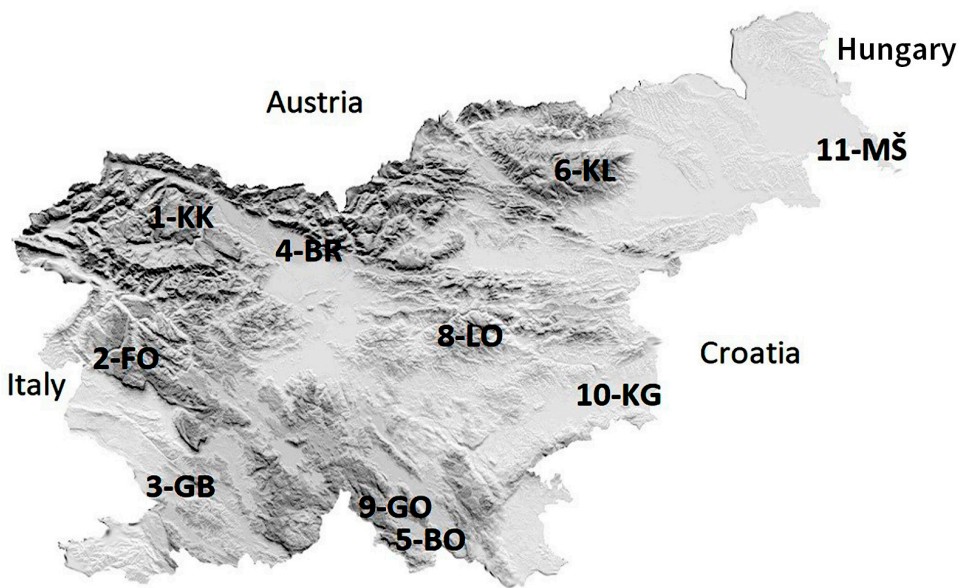

**Figure 1.** Locations of the Intensive Monitoring (IM) sites in Slovenia: 1-KK—Krucmanove konte, 2-FO—Fondek, 3-GB—Gropajski bori, 4-BR—Brdo, 5-BO—Borovec, 6-KL—Kladje, 8-LO—Lontovž, 9-GO—Gorica, 10-KG—Krakovski gozd, 11-MŠ—Murska šuma.

Until 2004, forests in the IM sites were managed mostly in a less intensive way (such as selective cutting), and forest stands on all IM sites were relatively closed without canopy gaps. In general, the studied forests were uneven-aged with different vertical vegetation layers.

Within each selected IM site, a relatively homogenous monitoring area ranging from 1 to 3 hectares was selected. In accordance with the standard protocol of the IM programme [23,24], 4 or 8 plots of 10 × 10 m were placed systematically across the area (Table 1), with a minimum distance between plots of 20 metres. The number of plots and the total size of monitoring area per site depend on the intensity and type of monitoring activities. On IM sites with smaller monitoring area (1 ha), 4 plots were placed, and monitoring activities were less intense. On sites with larger monitoring area (2 to 3 ha), 8 plots were installed, and more intense monitoring was conducted. These sites were additionally equipped with sampling devices for monitoring of different variables, e.g., atmospheric deposition, soil, and soil solution chemistry, meteorology, tree growth, and litterfall. In total, 60 plots were placed across all IM sites.

### 2.2. Vegetation Survey

A comprehensive survey of vascular plant species across the 60 plots in the 10 IM sites was performed in the summer in two monitoring periods (P1, P2); P1: years 2004 and 2005 and P2: years 2014 and 2015. On all IM sites, the period between the two surveys was ten years. The position of the 60 plots did not change between the two monitoring periods; the exact same area was re-surveyed in the second monitoring period (P2). In both monitoring periods, all identification of plant species, estimation of plant cover and other records were done by the same observer, i.e., first author of this paper, with field assistance of colleagues.

A visual estimation of total plant cover was made in the following vertical layers [24]:

- Tree layer: including woody plants and all climbers; height > 5 m;
- Shrub layer: including only woody plants and all climbers; height between 0.5 and 5 m;
- Herb layer: including all non-woody plants, and woody plants with height ≤ 0.5 m, including tree seedlings and browsed trees;
- Moss layer: separate record of mosses on the forest floor (on mineral soil) and mosses on other substrata, e.g., deadwood, living wood, rocks.

Total vegetation cover was defined as a vertical projection of all vegetation layers (including moss layers) on the ground. A separate record for each vascular plant species occurring in the herb, shrub and tree layer was performed in both surveys (P1 and P2). Tree species in lower- and upper-tree layers were recorded separately. The height of the lower-tree layer was defined as 75% of the maximum tree height in a stand, while stems in the upper-tree layer were above this threshold.

The visual estimates of plant species cover were done using a modified Barkman's method [35]. The vertical projection of plants' living parts on the floor was estimated in percent cover. A visual estimate of the plot surface covered by rocks and dead-wood material was assessed in both periods.

The nomenclature for vascular plants followed the Flora Europaea [36,37] and National Flora [38].

## 2.3. Disturbance Events and Evaluation

The disturbance regime across different Slovenian forest types is complex, with considerable variation in disturbance agents, sizes, and severities [39]. Due to several anthropogenic land uses and more settlements at lower altitudes, lowland forests in this area were more affected in the past and are still under pressure from several degradative processes, such as intensified land use, fragmentation, and pollution [40–42]. In recent times, floodplain and other lowland forests are increasingly exposed to invasion of alien plant species [43,44]. A variety of disturbance agents, such as windstorms, ice storms, wet snow, and insect outbreaks, caused marked damage to forests in Slovenia in the last decades [39]. In January–February 2014, an extreme ice storm caused damage to more than half a million ha of forests across Slovenia and Croatia. In terms of size and total wood volume damaged, this event was the most catastrophic natural disturbance on record for this region [39,45]. The ice storm and severe bark beetle outbreaks (*Ips typographus* L.) that followed [46–48] severely damaged or devastated large areas of forests.

In 2004, less than 10% of stands and upper-soil layers in all IM sites and plots were significantly disturbed. Between the two monitoring periods, disturbance events and intensity were monitored across the IM sites and plots (Table 2).

In addition to the disturbances listed in Table 2, most of the studied sites and plots were also exposed to significant deer browsing, which affects the structure and composition of natural regeneration [49]. Natural regeneration is significantly influenced by browsing across 52% of Slovenian forest [50]. Some of the IM sites, such as Fondek (2), Borovec (5), and Gorica (9), occur in regions with the highest impact of browsing. The Krucmanove konte site (1) on Pokljuka plateau was also impacted by occasional livestock grazing during the summer.

At the forest stand level, all disturbance impacts were estimated at each plot and its buffer zone. The buffer zone of a plot was situated around the periphery of the studied 100 m$^2$ plot area, and the distance from the plot margin to the outer margin of a buffer zone was 5 m. The estimate of disturbance impacts is an integral measure of natural and/or anthropogenic disturbances that caused damage to some component of the stand, including mortality of entire trees, damage to tree crowns, salvage logging of damaged or dead trees, regular tree cutting prescribed by forest management plan. This estimation also consisted of disturbances related to ground-surface layers, including the damage of ground-vegetation layers and upper-soil layers, e.g., tree uprooting and soil perturbations caused by felling and skidding operations. Deer browsing was not detected by this method. On each plot and its buffer zone, the proportion of forest stand and area of ground-surface layer significantly damaged from different disturbance factors was visually estimated in the percentage scale (%), ranging from 0 to 100%.

**Table 2.** Major disturbance events in the Intensive Monitoring sites between 2004 and 2015.

| IM Site | Disturbances in 1st Half of Period | Disturbances in 2nd Half of Period |
|---|---|---|
| 1. KRUCMANOVE KONTE | Windthrows (2005, 2007, 2009, 2015)—mostly outside the plots; Intensive logging activities after all windthrows—mostly outside the plots | Windthrows (y. 2015)—mostly outside the plots; Intensive logging activities after all windthrows—mostly outside the plots |
| 2. FONDEK | | Windthrow and logging of a few trees (2011) |
| 3. GROPAJSKI BORI | Mortality of single trees of *Pinus nigra* (all years) due to different fungal diseases and various insects | Mortality of single trees of *Pinus nigra* (all years); Windthrow and logging of a few trees (2010) |
| 4. BRDO | | Snow-break of trees (2010), Windthrow (2012); Ice storm (2014) and logging after disturbances; Waterpipe-system reconstruction across plots area (2013/2014) |
| 5. BOROVEC | Windthrow (2008)—mostly outside the plots; | Ice storm damaged parts of tree crowns (2014), logging activities after disturbances—mostly outside the plots |
| 6. KLADJE | Windthrows damaged a few trees (2005, 2006, 2009)—all outside the plots | Windthrows damaged a few trees (2014)—all outside the plots |
| 8. LONTOVŽ | | Regular selective management (2012)—party in plots, but mostly outside |
| 9. GORICA | Logging activities and forest-road construction near the plots (2008/2009) | Ice storm (2014) and intensive logging activities in all plots and in a wider area (2014/2015) |
| 10. KRAKOVSKI GOZD | Mortality of *Fraxinus angustifolia* trees (all years), mainly caused by altered flooding regime (soil drying because of lower groundwater level) and fungal infections (e.g., ash dieback) | Mortality of *Fraxinus angustifolia* trees (all years); Main logging activities in 2014 and 2015 |
| 11. MURSKA ŠUMA | Extreme flooding (2005, 2009), Intensive logging activities (2008) | Extreme flooding (2013) |

*2.4. Data Analysis*

We calculated the number of vascular plant species (*N*) across the 60 plots, including measures of diversity; E = evenness (equitability); and H' = Shannon diversity index. We tested the differences in vegetation-layer cover and diversity indices between the two monitoring periods (P1 and P2) with the Wilcoxon matched pairs test. Tests were carried out using Statistica 64 software (Dell Inc., Round Rock, TX, USA) [51].

For the entire plot-by-species matrix, we performed an unconstrained ordination Principal Coordinates Analysis (PCoA) using the 'pca' function in the *labdsv* package [52] for R software [53]. The input distance matrix was constructed based on Bray-Curtis dissimilarity index, using function 'vegdist' in the *vegan* package [54]. Abundance data were log-transformed prior to analysis, using $\log(x+1)$ transformation. To quantify the magnitude of compositional change for each plot, Euclidean distance was calculated based on sample scores for the first two PCoA axes, as they explained the highest proportion of compositional variability. For instance, Euclidean distance equal to 0 means that there was no change in species composition between P1 and P2, i.e., the position of the sample (plot) did not change in the PCoA ordination space. To test the effect of disturbance on temporal change in vascular plant composition, a simple linear regression was performed based on the averaged site-level data.

## 3. Results

### 3.1. Plant Diversity

The total number of vascular plant species decreased from 272 to 243 species (10.0% decrease) during the duration of the study across the 10 sites. Significant species turn-over was documented; 46 species identified in P1 (16.9% of all species in P1) were not found in P2, and 17 new species (not present in P1) were recorded in P2. On average, the frequency of species occurrence increased from 6.9 plots in P1 up to 7.4 plots in P2.

Although we found a decrease in plant diversity pooled across all the study sites, the mean number of plant species per site did not differ significantly between P1 and P2 (Table 3). However, site-scale values of diversity indexes (E and H') significantly increased (Table 3, Figure 2). We also observed a significant decrease of tree-layer cover (Table 3).

**Table 3.** Comparison of site-scale variables between the two monitoring periods (P1: years 2004/2005, P2: years 2014/2015), including results of the Wilcoxon Matched Pairs Test.

|  | Period 1 (2004/2005) | Period 2 (2014/2015) |  |  |
| --- | --- | --- | --- | --- |
|  | Mean (Cover in %) | Mean (Cover in %) | Z | Significance Level |
|  | (*n* = 10) | (*n* = 10) |  |  |
| Deadwood cover | 4.4 | 5.9 | 0.291 | ns. |
| Total vegetation cover | 99.3 | 96.3 | 0.775 | * |
| Tree-layer cover | 84.1 | 78.4 | 0.787 | * |
| Shrub-layer cover | 7.7 | 9.3 | 0.032 | ns. |
| Herb-layer cover | 72.9 | 71.5 | 0.355 | ns. |
| Ground moss cover | 4.0 | 3.6 | 0.085 | ns. |
| Other moss cover | 1.7 | 1.8 | 0.465 | ns. |
| N of species | 37.7 | 37.5 | 0.048 | ns. |
| Evenness E | 0.55 | 0.60 | 0.838 | ** |
| Shannon H' | 1.94 | 2.12 | 0.838 | ** |

Significance level: *** = $p < 0.001$; ** = $0.001 < p < 0.010$; * = $0.010 < p < 0.050$, ns. = $p > 0.050$.

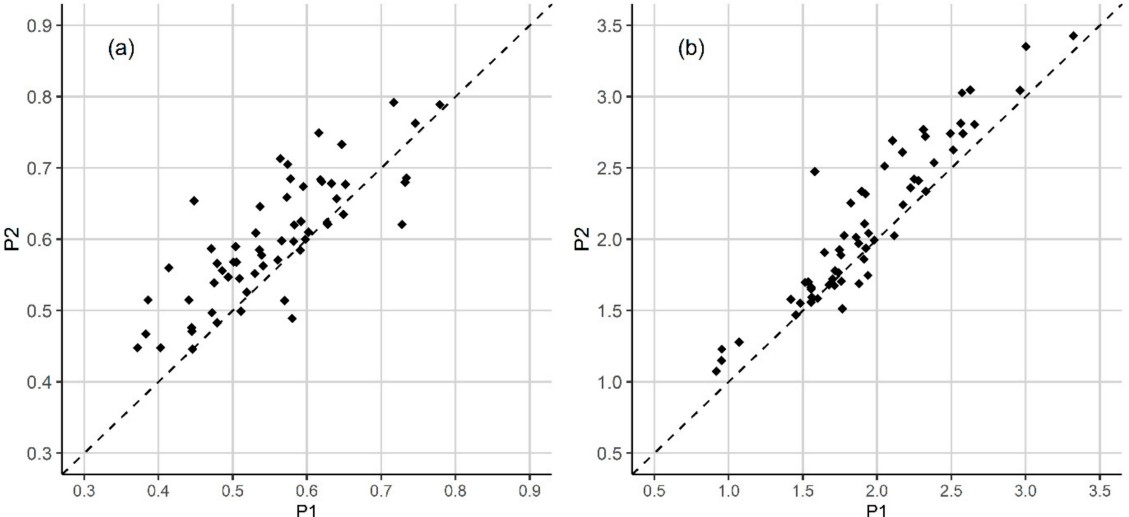

**Figure 2.** Changes in (**a**) Evenness—E and (**b**) Shannon diversity index—H' between the two monitoring periods; P1: years 2004/2005, P2: years 2014/2015. Each data point indicates a sampling plot. Points above the dashed line (slope = 1) indicate plots with increased E (H') value, and points under the line indicate plots with a decreased E (H') value from period P1 to P2.

The frequency and cover of most common plant species changed during the study duration (Table 4). For example, *Fagus sylvatica* significantly decreased in the upper-tree layer but increased in the herb layer. The only significant change in the shrub layer was that of *Acer campestre* L., which increased in cover over time. In the herb layer, several species significantly increased in cover, including *Anemone nemorosa* L., *Viola reichenbachiana* Jord. ex Boreau, *Cyclamen purpurascens* Mill., *Hedera helix* L., *Daphne mezereum* L., *Carex sylvatica* Huds., *Galeobdolon flavidum* (L.) Crantz, *Salvia glutinosa* L., and *Brachypodium sylvaticum* (Huds.) Beauv. (Table 4). Changes in frequency were variable; 11 species were present in the same number of plots in both periods, 13 species increased, and 14 species decreased in frequency (Table 4). It is important to note that although many common species decreased in frequency, most species increased in cover during the study period (Table 4, Figure 3).

**Table 4.** Frequency and cover of common plant species in the two monitoring periods; P1: years 2004/2005, P2: years 2014/2015. Differences in the mean cover of plant species between the two monitoring periods was tested by the Wilcoxon Matched Pairs Test. Species with occurrence on 25% or more of the plots in at least one period are shown.

| PLANT SPECIES | VERTICAL LAYER | N plots (P1) | N plots (P2) | COVER (P1) | COVER (P2) | COV. DIFF. (P2-P1) | Z | Significance Level |
|---|---|---|---|---|---|---|---|---|
| *Fagus sylvatica* L. | UT | 28 | 28 | 34.4 | 31.9 | −2.5 | 2.310 | * |
| *Fagus sylvatica* L. | LT | 25 | 27 | 5.5 | 4.9 | −0.6 | 1.014 | ns. |
| *Daphne mezereum* L. | S | 30 | 25 | 0.3 | 0.2 | −0.1 | 1.193 | ns. |
| *Acer pseudoplatanus* L. | S | 24 | 24 | 0.4 | 0.9 | 0.5 | 0.698 | ns. |
| *Fagus sylvatica* L. | S | 24 | 24 | 2.0 | 1.8 | −0.2 | 0.876 | ns. |
| *Acer campestre* L. | S | 17 | 16 | 0.4 | 0.9 | 0.5 | 2.845 | ** |
| *Acer pseudoplatanus* L. | H | 34 | 35 | 1.6 | 2.1 | 0.5 | 1.870 | ns. |
| *Anemone nemorosa* L. | H | 29 | 29 | 0.4 | 0.7 | 0.3 | 2.636 | ** |
| *Fagus sylvatica* L. | H | 28 | 29 | 1.0 | 1.8 | 0.8 | 2.490 | * |
| *Viola reichenbachiana* Jord. ex Boreau | H | 28 | 29 | 0.1 | 0.3 | 0.2 | 2.542 | * |
| *Cyclamen purpurascens* Mill. | H | 27 | 28 | 0.8 | 1.6 | 0.8 | 3.432 | *** |
| *Mercurialis perennis* L. | H | 27 | 27 | 2.0 | 2.1 | 0.1 | 0.734 | ns. |
| *Asarum europaeum* L. | H | 25 | 25 | 0.7 | 0.8 | 0.1 | 0.621 | ns. |
| *Hedera helix* L. | H | 24 | 25 | 0.4 | 0.5 | 0.1 | 2.310 | * |
| *Polygonatum multiflorum* (L.) All. | H | 26 | 23 | 0.2 | 0.2 | 0.0 | 0.201 | ns. |
| *Cardamine trifolia* L. | H | 24 | 24 | 0.7 | 0.7 | 0.0 | 0.135 | ns. |
| *Daphne mezereum* L. | H | 22 | 26 | 0.2 | 0.3 | 0.1 | 2.329 | * |
| *Galium odoratum* (L.) Scop. | H | 25 | 23 | 0.7 | 0.9 | 0.2 | 0.971 | ns. |
| *Carex sylvatica* Huds. | H | 23 | 23 | 0.1 | 0.3 | 0.1 | 2.511 | * |
| *Oxalis acetosella* L. | H | 21 | 22 | 0.5 | 0.5 | 0.0 | 1.826 | ns. |
| *Galeobdolon flavidum* (L.) Crantz | H | 21 | 21 | 0.5 | 0.8 | 0.3 | 2.701 | ** |
| *Rosa arvensis* Huds. | H | 21 | 20 | 0.4 | 0.4 | 0.0 | 0.059 | ns. |
| *Aremonia agrimonoides* (L.) DC. | H | 20 | 20 | 0.2 | 0.2 | 0.0 | 0.114 | ns. |
| *Picea abies* (L.) H. Karst. | H | 21 | 19 | 0.4 | 0.4 | 0.0 | 0.210 | ns. |
| *Mycelis muralis* Dumort | H | 21 | 18 | 0.2 | 0.2 | 0.0 | 0.489 | ns. |
| *Solidago virgaurea* L. | H | 19 | 20 | 0.1 | 0.1 | 0.0 | 0.853 | ns. |
| *Carex digitata* L. | H | 19 | 19 | 0.3 | 0.3 | 0.0 | 1.098 | ns. |
| *Euphorbia amygdaloides* L. | H | 19 | 19 | 0.3 | 0.4 | 0.1 | 1.604 | ns. |
| *Fragaria vesca* L. | H | 19 | 18 | 0.2 | 0.2 | 0.0 | 0.357 | ns. |
| *Salvia glutinosa* L. | H | 19 | 18 | 0.3 | 0.5 | 0.2 | 2.192 | * |
| *Acer campestre* L. | H | 17 | 19 | 0.4 | 0.6 | 0.2 | 1.922 | ns. |
| *Brachypodium sylvaticum* (Huds.) Beauv. | H | 17 | 19 | 0.1 | 0.3 | 0.2 | 2.585 | ** |
| *Dryopteris filix-mas* (L.) Schott | H | 18 | 17 | 0.3 | 0.3 | 0.0 | 0.535 | ns. |
| *Sorbus aucuparia* L. | H | 16 | 19 | 0.1 | 0.2 | 0.1 | 1.922 | ns. |
| *Sanicula europaea* L. | H | 19 | 15 | 0.5 | 0.5 | 0.0 | 0.153 | ns. |
| *Lathyrus vernus* L.) Bernh. subsp. *vernus* | H | 16 | 17 | 0.2 | 0.2 | 0.0 | 0.296 | ns. |
| *Senecio fuchsii* Gmelin | H | 16 | 17 | 0.2 | 0.3 | 0.1 | 0.612 | ns. |
| *Hacquetia epipactis* (Scop.) DC. | H | 17 | 15 | 1.1 | 1.1 | 0.0 | 0.000 | ns. |
| *Gentiana asclepiadea* L. | H | 16 | 15 | 0.2 | 0.2 | 0.0 | 0.730 | ns. |
| *Omphalodes verna* Moench | H | 16 | 15 | 1.0 | 1.1 | 0.131 | 0.676 | ns. |

Legend: Vertical layers: UT—upper-tree layer, LT—lower-tree layer, S—shrub layer, H—herb layer, Significance level: *** = $p < 0.001$; ** = $0.001 < p < 0.010$; * = $0.010 < p < 0.050$, ns. = $p > 0.050$.

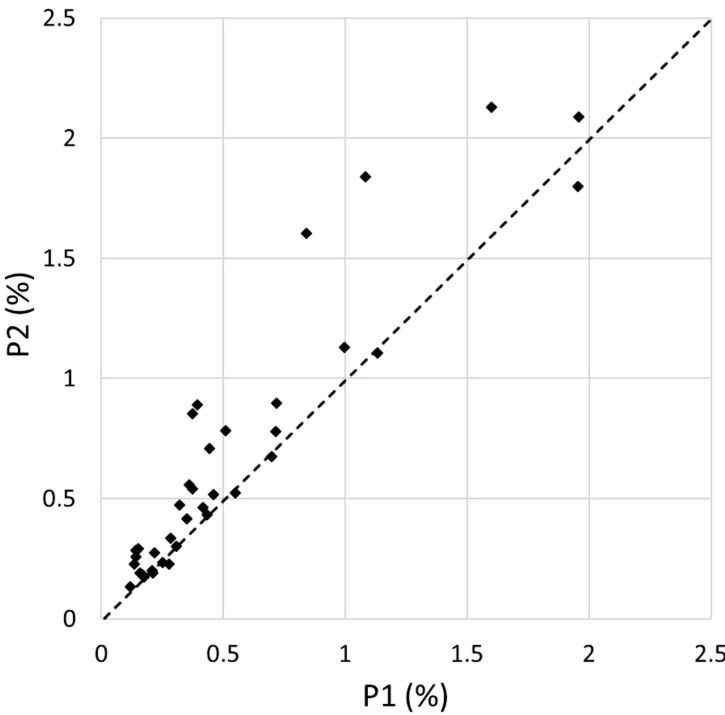

**Figure 3.** Mean cover (in %) of the most common understory plant species (in herb and shrub layer) in two monitoring periods; P1: years 2004/2005, P2: years 2014/2015. Dots indicate 38 plant species that were present at least in 25% of all plots (see Table 4).

*3.2. Disturbance Impacts*

In P1 there were very few significant disturbances recorded on plots. The IM sites and plots selected in 2004 were purposely established in areas without some noticeable disturbances. Consequently, the estimation of disturbance impacts on all plots and IM sites was low (on average less than 1%) and ranged from 0% to 10% per plot. However, during the 10-year study period, especially in the second half, disturbance increased across most of the IM sites (Table 2). At the end of this period, the estimates of disturbance impacts ranged from 0% to 40% per plot, and its overall mean value per site was 16.3%. It increased significantly (Wilcoxon matched pairs test: $p = 0.006$), with more than 10% of stand and/or ground layer disturbance on 40 of 60 plots in P2. On average, the estimated disturbance in P2 was the highest on plots in the Gorica site (mean value=38.8%), followed by Murska šuma (26.3%), Brdo (19.4%) and Krakovski gozd (18.8%). Very few disturbances were documented at the Kladje site (5.0%) (Table 5).

**Table 5.** Estimated disturbances in P1 (2004/2005) and P2 (2014/2015) for each IM site.

| | Period 1 (2004/2005) | Period 2 (2014/2015) |
|---|---|---|
| **IM Ste** | **Mean (%) ± SE** | **Mean (%) ± SE** |
| 1. KRUCMANOVE KONTE (1-KK) | 2.5 ± 2.5 | 16.3 ± 3.2 |
| 2. FONDEK (2-FO) | 0.6 ± 0.6 | 9.4 ± 2.1 |
| 3. GROPAJSKI BORI (3-GB) | 0.0 ± 0.0 | 13.8 ± 4.3 |
| 4. BRDO (4-BR) | 1.3 ± 0.8 | 19.4 ± 2.6 |
| 5. BOROVEC (5-BO) | 0.6 ± 0.6 | 12.5 ± 0.9 |
| 6. KLADJE (6-KL) | 0.0 ± 0.0 | 5.0 ± 3.5 |
| 8. LONTOVŽ (8-LO) | 0.0 ± 0.0 | 8.8 ± 3.2 |
| 9. GORICA (9-GO) | 2.5 ± 2.5 | 38.8 ± 1.3 |
| 10. KRAKOVSKI GOZD (10-KG) | 1.3 ± 1.3 | 18.8 ± 3.8 |
| 11. MURSKA ŠUMA (11-MŠ) | 0.6 ± 0.6 | 26.3 ± 3.4 |

Although absolute changes in species composition between P1 and P2 were rather small, we found a strong positive relationship between disturbance and the magnitude of compositional shifts ($R^2 = 0.719$, $p = 0.002$). IM sites with more intense disturbances exhibited larger changes in composition and vice-versa (Figure 4).

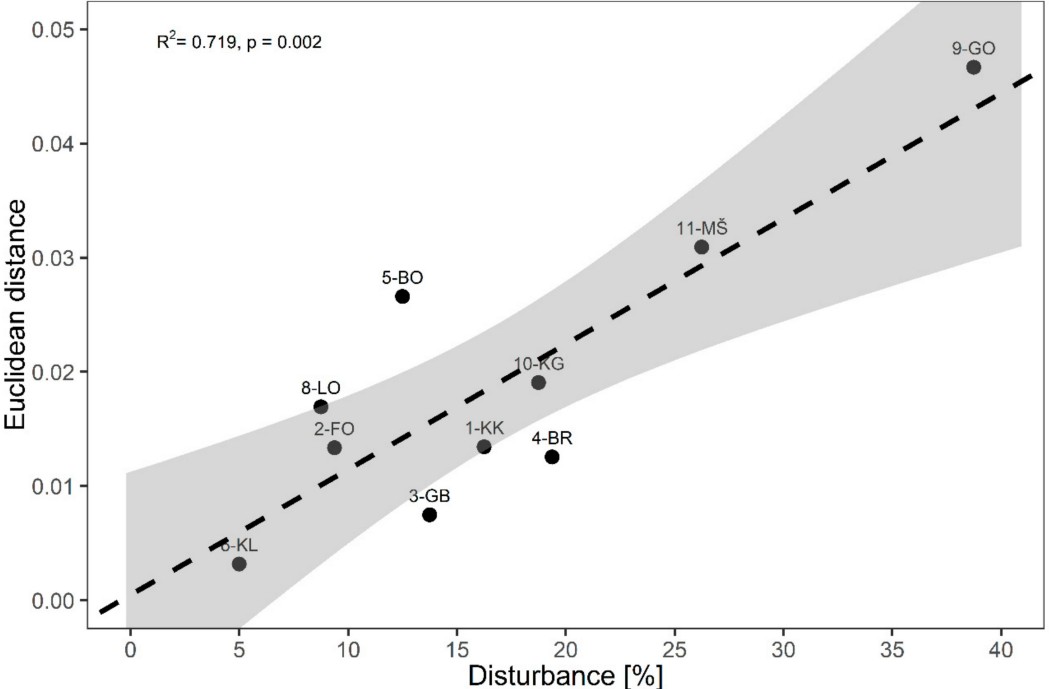

**Figure 4.** Linear relationship between site-level disturbance (%), estimated in P2, and the degree of compositional changes in forest vegetation, which was measured as Euclidean distance in PCoA ordination space. The grey band around the regression line denotes a 95% confidence interval. IM site labels correspond to Table 1.

## 4. Discussion

In forest ecosystems studied within the ICP Forests monitoring programme in Slovenia, a high level of plant diversity was documented at the beginning of monitoring period [30–32]. Over the past decade, there has been substantial species turnover and a decrease in gamma diversity across the network of sites. This could be interpreted in the context of accelerating biodiversity losses worldwide and concerns about species impoverishment [55,56]. At the site scale, however, Shannon diversity index and evenness increased over time. Species-level responses revealed that the most frequent or common species (habitat generalists) across the sites largely increased in cover over time, which may have led to loss of more rare or infrequent species (e.g., habitat specialists).

Observed changes in forest vegetation were clearly driven by various disturbance events, spanning from local tree- or ground-layer disturbances, more or less limited to the investigated plot area, as well as more widespread disturbances (e.g., gradual canopy mortality) inducing profound alterations to stand canopy structure in the surroundings of our study sites. A distinct change in the network of IM sites across Slovenia was the transition from homogenous, closed canopy stands to more open stands due to a variety of local and widespread disturbance events that occurred during the past decade, including windthrow, snow-break, ice-storm damage, pathogens, bark-beetle outbreaks, and salvage logging activities that routinely follow natural disturbances [39,45–48,57]. Apart from quantified natural and anthropogenic disturbance effects, a variety of other processes could be directly or indirectly driving these diversity changes at different scales. Local and regional scale drivers, such as ungulate browsing damage [58] and legacies of past land use [59], have been shown to be responsible for detectable changes in understory composition over time [60]. Additionally, environmental factors and

disturbance agents operating on larger spatial and longer temporal scales, including large-scale climate change and atmospheric nutrient deposition [7,10] could contribute to plant community dynamics. Disentangling broad scale versus local scale drivers from our observational data is not possible, partly due to the relatively short monitoring period

Many of the plot and site scale changes in the understory vegetation documented here are likely a result of these recent disturbances, either by providing opportunities for early successional species (e.g., *Brachypodium sylvaticum*, *Carex sylvatica*, *Salvia glutinosa*), or by allowing expansion of existing competitive species due to increased understory light. The latter mechanism was probably widespread given that many of the common herb species increased in cover over time (e.g., *Anemone nemorosa, Viola reichenbachiana*, *Cyclamen purpurascens*, *Galeobdolon flavidum*). These species are considered as generalist species with wider ecological niches in comparison to the forest understory specialists. In most IM sites, most disturbances occurred in the overstory canopy, resulting in the reduction of tree-layer cover and consequent increases in light reaching the forest floor. As shown in many previous studies [61–63], typical forest understory species, presumably more sensitive to disturbance, are able to tolerate a wider range of light conditions.

Relatively small changes in species composition are likely because many of the disturbances across the network occurred in the second half of the monitoring decade, including the widespread and severe ice storm in 2014 that damaged forests across the country [39,45,64]. A longer monitoring period would be required to capture the understory response to these recent disturbances. Widespread severe bark-beetle disturbances and salvage logging followed the 2014 ice storm event [46–48,57], further opening the canopies at some of the IM sites, which is likely to lead to more pronounced changes to the understory in the future. Forest plant communities respond to changes of ecological conditions with some time lag. Species diversity is likely to increase in these forests as canopy gaps are colonized by new species. In previous and recent experimental studies that created gaps in the dominant type of beech forests in Central Europe and Slovenia, a high number of early successional species colonized gaps [65,66]. However, disturbances can have direct negative effects on forest vegetation. For example, logging and skidding trails on many plots and waterpipe-system reconstruction across the Brdo IM site caused severe soil damage and understory plant removal. Furthermore, increases in diversity from post-disturbance succession could be offset by community homogenization due to invasive species [44] and nitrogen deposition [60].

Beside community-level changes driven by disturbances, the decreased number of species (gamma diversity), higher frequency of species occurrence per plot, and increase in evenness at the end of monitoring period might be, at least to some extent, interpreted as homogenization of vegetation [67–69], although the short monitoring period and restricted region may not enable us to distinguish such non-random changes in the forest understory plant communities, which were recently reported in long-term studies across Europe [70], and references therein]. Nevertheless, the observed species- and community-level changes could be an early sign that understory vegetation is becoming floristically more similar in different forest types. Further monitoring is needed to confirm such assumptions as our study addresses relatively short-term changes of the understory vegetation.

Determining the short-term effects of forest management on forest biodiversity is critical for effective conservation of forest biodiversity. It is important to have knowledge on the recovery capacity of a given forest type and the vulnerability of the vegetation—the degree of difficulty of recovering from the effects of either a natural or anthropogenic disturbance; the slower and/or more difficult the recovery process, the more vulnerable the habitat type is [71]. This means that it is important to know the rate at which the forest recovers to its initial state after cessation of the land use to which it was subjected. The habitats that contain mainly late successional vegetation, such as most of the forests in this study, require more time to return to their original state after natural or anthropogenic disturbances [71].

Across Slovenian forests, sustainable close-to-nature forest management (i.e., continuous cover forestry utilizing small-scale uneven-aged silviculture) has been traditionally favoured over intensive,

even-aged management principles. However, it is not well known to what extent close-to-nature silvicultural systems influence plant diversity in forest ecosystems [6]. Other authors have noted that this type of management may harbour fewer plant species than even-aged (rotation) forestry [72]. The same authors conclude that there is no evidence that sustainable forest management has led to decreased biodiversity in Central Europe [72]. However, even in sustainable forest management using a small-scale uneven-aged approach, frequent disturbance events are likely to lead to a higher number of plant species colonizing open areas, particularly during early successional phases.

In addition to the marked increase in disturbance over the past decade [45–48], climate in the studied region has changed considerably in recent decades [73], particularly in the form of increased mean temperatures and more frequent and extended heat waves. Future climate change is likely to amplify the forthcoming development and interact with natural disturbance regimes in the coming decades. These changes are expected to interrupt the sustainable provision of forest ecosystem services to society [16] while plant species diversity may increase in the short-term. Since the present study addresses a relatively short-term perspective, more reliable evaluation of intensified disturbance regime impacts on vegetation across Slovenian forests will be possible with future monitoring in the studied IM sites.

## 5. Conclusions

Our results provided evidence that a wide range of disturbance types and their impacts triggering an increase in the magnitude of compositional shifts were the dominant processes shaping forest understory communities during the 10-year monitoring period. These findings warrant further research questions regarding the role of disturbance as a mechanism for the future development of understory communities. Beside documented disturbances in the period from 2004/05 to 2014/15, frequency and intensity of disturbances in the next decades are predicted to increase [56], with decisive consequences for understory species composition and diversity. Forest floor vegetation is tightly linked to characteristics of the overstory (tree layer cover and composition), mainly through its effects on understory light regime [56,70]. Natural disturbances slowly reducing or completely removing tree canopy cover, together with small-scale disturbances, including forest management interventions, will put the buffering capacity of forest stands to a serious test and possibly provoke more rapid changes of the forest understory. The interplay between past and present/future disturbances, with potential cascading effects, should thus deserve special attention from forest managers and conservationists. In order to successfully address such contemporary issues, the established systematic monitoring on IM sites serve as a strong foundation for more reliable quantification of these effects and can importantly contribute to the long-term forest vegetation resurvey data in this part of Europe. The compositional changes observed in our study can be partly attributed to natural dynamics of forest vegetation and to effects of intensified forest disturbances. Future work will thus be dedicated to re-surveys of the study plots. Furthermore, additional analysis addressing variation between IM sites and the recognition of which sites (forest types) have more stable vegetation composition and which are more subjected to changes would be insightful. Lastly, utilization of different approaches (e.g., plant functional traits, ecological indicator values) will likely deepen our understanding of potential factors behind temporal vegetation changes across temperate forest ecosystems in Slovenia.

**Author Contributions:** The conceptualization was done by L.K.; preparation of methodology by L.K., T.A.N., J.K.; formal analysis by L.K. and J.K.; resources by L.K.; data curation by L.K. and J.K.; writing—original draft preparation by L.K., T.A.N., J.K.; funding acquisition by L.K.

**Funding:** The study was funded by Slovenian Ministry of Agriculture, Forestry and Food, and Slovenian Research Agency (research core funding No. P4-0107).

**Acknowledgments:** The study was performed as a part of the Intensive Monitoring Programme in Slovenia (part of EU Programme ICP-Forests). We would like to thank Matej Rupel, Aleksander Marinšek, Daniel Žlindra, Primož Simončič and other colleagues from the Slovenian Forestry Institute for different ways of assistance, and many experts from the Slovenian Forest Service for collecting field data on disturbances and management. We thank to anonymous reviewers for their constructive criticism and valuable suggestions.

**Conflicts of Interest:** The authors declare no conflict of interest. The funders had no role in the design of the study; in the collection, analyses, or interpretation of data; in the writing of the manuscript, or in the decision to publish the results.

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
