# Peer review of "Effects of Disturbance on Understory Vegetation across Slovenian Forest Ecosystems"

_forests, doi:10.3390/f10111048_

Round 1

Reviewer 1 Report

I think this updated version of the manuscript is much improved over the last version. The authors put together a more careful approach to data analysis and interpretation of results. The one remaining major concern with the manuscript is the focus on “biotic homogenization” and “compositional convergence” of vegetation that is described in the Discussion (P11 L283-292) and again in the Conclusion (P12 L321-325). There is no empirical evidence for these trends in vegetation apart from the measures of diversity and evenness among study sites and plots. However, these statistics do not provide enough information to make such a conclusion. A more prudent approach to address "biotic homogenization" would be to analyse the data using multivariate statistics and compare statistically the sites in terms of vegetation composition between P1 and P2, but this was not done. Concluding that vegetation is being homogenized or converging is speculative and unsupported by the data. The main point to be delivered for the manuscript is that a wide range of disturbances were documented and quantified among study sites, and these were related to changes in vegetation composition among sites, with implications for future vegetation development across sites. Namely, the larger the disturbance within a site the greater was the observed change in vegetation composition (Figure 4). These are powerful enough conclusions without having to speculate on factors that were not quantified such as “biotic homogenization” and “compositional convergence”. Focus on the findings that are supported by data only.

Minor comments to be addressed:

Abstract. P1 L21. “ranged from 6.0% to 38.8% per site in 2014/15” but what was the range for 2004/5?

Abstract. P1 L21. Change “corresponded with a reduction of total vegetation cover”, to “corresponded to a reduction in total vegetation cover”.

Introduction. P2 L49. The following statement needs a citation: “Damage from wind, insect outbreaks and wildfires have increased in Europe’s forests throughout the twentieth century”.

Methods. P 2 L86-90. Explain in more detail why the monitoring areas ranged from 1 to 3 hectares. Also, explain in more detail the number of plots that were placed at each site in relation to intensity and type of monitoring activities. These factors influence interpretation of results and should be described.

Methods. P4 L116. Should this statement read, “The height of the lower-tree layer was defined as trees that were less than 75% of the maximum tree height in a stand”?

Methods. P4 L121-123. Move this statement (“In both monitoring periods…”) after the one ending “the second monitoring periods (P2)” on line 105.

Methods. P5 L137. What is the species of bark beetle?

Table 2. The table needs line divisions between IM sites in order to relate the text to sites.

Table 2. Site 10. What were the agents of mortality for Pinus trees for site 3, and Fraxinus trees for site 10? Providing these details in the methods would be appropriate.

Methods. P6 L150. How were “significant disturbance impacts” determined? Why only significant disturbance impacts and not all disturbances?

Methods. P6 L152. What is the area of the buffer zone? The buffer zone does not appear to be included in analyses or mentioned in the results or discussion sections of the manuscript and should probably be excluded from the manuscript altogether.

Methods. P6 L157-158. How were disturbances such as tree tip-up mounds or harvested trees estimated in terms of cover measures?

Methods. P6 L165-166. What was the measure of species resemblance for the PCoA? Euclidean?

Methods. P6 L167. How were species abundance values log transformed for analysis? As ln(x+1) where x is the original abundance value for a species? You cannot take a log of zero values.

Results. P6 L181-182. “Although we found a decrease in plant diversity pooled across all the study sites”. What was this decrease among all sites? Also, “mean number of plant species per site not differ significantly between P1 and P2.” What are these values?

Table 4. Number of plots are shown but I assume the Wilcoxon Matched Pairs tests were done at the site-level (N=10) to avoid pseudoreplication?

Discussion. P10 L245. “follows a common global pattern of increasing rate of global biodiversity loss”. The results observed in this study are driven by disturbance (see Figure 4). The judgement that they conform to a pattern of global biodiversity loss is subjective given the variation in composition and diversity measures observed among study sites.

Discussion. P11. The discussion on this page could start with smaller scale forest processes such as disturbances first and then speculate on the effects of climate change and nutrient deposition later in the Discussion.

Discussion. P11. See comments at the top of this review where statements about “biotic homogenization” and “compositional convergence” of vegetation are unwarranted and should be removed. Focus on the main study findings that are backed up by study data.

Discussion and Conclusions. P11-12. What are the implications of the observed disturbances on the future development of forest vegetation?

End.

Author Response

Comments and Suggestions for Authors

I think this updated version of the manuscript is much improved over the last version. The authors put together a more careful approach to data analysis and interpretation of results. The one remaining major concern with the manuscript is the focus on “biotic homogenization” and “compositional convergence” of vegetation that is described in the Discussion (P11 L283-292) and again in the Conclusion (P12 L321-325). There is no empirical evidence for these trends in vegetation apart from the measures of diversity and evenness among study sites and plots. However, these statistics do not provide enough information to make such a conclusion. A more prudent approach to address "biotic homogenization" would be to analyse the data using multivariate statistics and compare statistically the sites in terms of vegetation composition between P1 and P2, but this was not done. Concluding that vegetation is being homogenized or converging is speculative and unsupported by the data. The main point to be delivered for the manuscript is that a wide range of disturbances were documented and quantified among study sites, and these were related to changes in vegetation composition among sites, with implications for future vegetation development across sites. Namely, the larger the disturbance within a site the greater was the observed change in vegetation composition (Figure 4). These are powerful enough conclusions without having to speculate on factors that were not quantified such as “biotic homogenization” and “compositional convergence”. Focus on the findings that are supported by data only.

ANSWER: We completely agree on this major comment. The interpretation of the results is now focused on disturbance-driven changes in the understory vegetation. The aspect of biotic homogenization has been reduced to a minimum and it is now only given as a hint at the end of the revised manuscript. Although we did not explicitly quantify the potential trend of compositional convergence, in our opinion, it is relevant to at least briefly mention this aspect in the Discussion. In the revised version, Discussion and Conclusions have been thoroughly reworked, according to given suggestions.

Minor comments to be addressed:

Abstract. P1 L21. “ranged from 6.0% to 38.8% per site in 2014/15” but what was the range for 2004/5?

ANSWER: OK, we added numbers for the period 2004/5.

Abstract. P1 L21. Change “corresponded with a reduction of total vegetation cover”, to “corresponded to a reduction in total vegetation cover”.

ANSWER: OK, we considered.

Introduction. P2 L49. The following statement needs a citation: “Damage from wind, insect outbreaks and wildfires have increased in Europe’s forests throughout the twentieth century”.

ANSWER: The citation for this statement is at the end of paragraph.

Methods. P 2 L86-90. Explain in more detail why the monitoring areas ranged from 1 to 3 hectares. Also, explain in more detail the number of plots that were placed at each site in relation to intensity and type of monitoring activities. These factors influence interpretation of results and should be described.

ANSWER: OK, we considered.

Methods. P4 L116. Should this statement read, “The height of the lower-tree layer was defined as trees that were less than 75% of the maximum tree height in a stand”?

ANSWER: Yes, you are correct.

Methods. P4 L121-123. Move this statement (“In both monitoring periods…”) after the one ending “the second monitoring periods (P2)” on line 105.

ANSWER: OK, we considered.

Methods. P5 L137. What is the species of bark beetle?

ANSWER: OK, we considered. It is European spruce bark beetle (Ips typographus).

Table 2. The table needs line divisions between IM sites in order to relate the text to sites.

ANSWER: OK, we considered.

Table 2. Site 10. What were the agents of mortality for Pinus trees for site 3, and Fraxinus trees for site 10? Providing these details in the methods would be appropriate.

ANSWER: OK, we considered. We added information on agents of mortality directly to Table 2.

Methods. P6 L150. How were “significant disturbance impacts” determined? Why only significant disturbance impacts and not all disturbances?

ANSWER: OK, we considered.

Methods. P6 L152. What is the area of the buffer zone? The buffer zone does not appear to be included in analyses or mentioned in the results or discussion sections of the manuscript and should probably be excluded from the manuscript altogether.

ANSWER: It is written in Methods that the disturbance impacts were estimated at each plot and its buffer zone too. The buffer zone is defined as a surrounding area of each plot, and it was situated around the periphery of the studied 100 m2 plot area. It is explained that the radius of buffer zone is 5 metres.

Methods. P6 L157-158. How were disturbances such as tree tip-up mounds or harvested trees estimated in terms of cover measures?

ANSWER: OK, we agree with you. These two types of disturbances were somehow overlooked in explanation. We added these two types in definition in Methods.

Methods. P6 L165-166. What was the measure of species resemblance for the PCoA? Euclidean?

ANSWER: OK, we considered. We provided additional information in the Methods section.

Methods. P6 L167. How were species abundance values log transformed for analysis? As ln(x+1) where x is the original abundance value for a species? You cannot take a log of zero values.

ANSWER: OK, we considered. We added additional sentence explaining this in the Methods.

Results. P6 L181-182. “Although we found a decrease in plant diversity pooled across all the study sites”. What was this decrease among all sites? Also, “mean number of plant species per site not differ significantly between P1 and P2.” What are these values?

ANSWER: OK, we considered. The decrease of number of species among all sites is explained in the first paragraph in Results as follows: “The total number of vascular plant species decreased from 272 to 243 species (10.0% decrease) during the duration of the study across the 10 sites.

The second part “mean number of plant species per site not differ significantly between P1 and P2” refers to Table 3. We did not show the numbers, because the difference is very small and non-significant.

Table 4. Number of plots are shown but I assume the Wilcoxon Matched Pairs tests were done at the site-level (N=10) to avoid pseudoreplication?

ANSWER: Yes, all statistical tests were done using site-level data (N=10).

Discussion. P10 L245. “follows a common global pattern of increasing rate of global biodiversity loss”. The results observed in this study are driven by disturbance (see Figure 4). The judgement that they conform to a pattern of global biodiversity loss is subjective given the variation in composition and diversity measures observed among study sites.

ANSWER: OK, we considered.

Discussion. P11. The discussion on this page could start with smaller scale forest processes such as disturbances first and then speculate on the effects of climate change and nutrient deposition later in the Discussion.

ANSWER: OK, we considered. This part was rewritten. In the second paragraph of Discussion, we started with the disturbance effects first, and climate change and nutrient deposition impacts are in the second part.

Discussion. P11. See comments at the top of this review where statements about “biotic homogenization” and “compositional convergence” of vegetation are unwarranted and should be removed. Focus on the main study findings that are backed up by study data.

ANSWER: OK, we considered. These parts were rewritten or even removed.

Discussion and Conclusions. P11-12. What are the implications of the observed disturbances on the future development of forest vegetation?

ANSWER: OK, we considered. We added more discussion on the implications of the observed disturbances on future forest development. This is now addressed (directly and indirectly) at several places in the Conclusions, also some parts in Discussion.

Reviewer 2 Report

No comment

Author Response

Comments and Suggestions for Authors

No comment

ANSWER: OK, there was nothing to considered or rewritten.

This manuscript is a resubmission of an earlier submission. The following is a list of the peer review reports and author responses from that submission.

Round 1

Reviewer 1 Report

I do not see, why you need two aims, at least in relation to study title. The first aim sounds more as objective.

How you deal with the edge effect, how you distinguish it between the effects of disturbances.

On lines136-138 you mentioned significant deer browsing impact on forest natural regeneration, how does this has taken into account in the assessment of Stand Disturbance Index.

Lines 239-245 does in the correlation of SDI and different structural and diversity variables has been considered site specific impacts?

Conclusions is missing, but what is the take home message? what are the the direction of changes.

Author Response

Manuscript ID: forests-583328

Title: Effects of disturbance on understory vegetation across Slovenian forest ecosystems

Authors: Lado Kutnar *, Thomas Andrew Nagel

Received: 13 August 2019

E-mails: lado.kutnar@gozdis.si, tom.nagel@bf.uni-lj.si Submitted to section: Forest Ecology and Management, https://www.mdpi.com/journal/forests/sections/Ecology_Management

Instruction for authors: 

Ø Authority on Latin binomial should be provided after each common name the first time referred to in the text.

# We added the authors of all Latin binomial species names addressed in our manuscript. Authority of Latin binomial names were provided in the text, and in Table 4 where the most common species are listed.

Ø The number of significant digits should be based on the precision of the analytical method and be rounded accordingly, and the variables presented should be correct and consistent.

# The methods and evaluation of variables suggested by the editors and reviewers were significantly improved. The number of digits in many cases were reduced according to the method used.

Ø Conclusions: this section is mandatory and should be added to the manuscript.

# OK, we added a Conclusion section to summarize the main focus of the study and suggestions for further, long-term studies.

------------------------------------------

Your co-authors can also view this link if they have an account in our submission system using the e-mail address in this message.

Please revise the manuscript according to the reviewers' comments and upload the revised file within 10 days. Use the version of your manuscript found at the above link for your revisions, as the editorial office may have made formatting changes to your original submission. Any revisions should be clearly highlighted, for example using the "Track Changes" function in Microsoft Word, so that changes are easily visible to the editors and reviewers. Please provide a cover letter to explain point-by-point the details of the revisions in the manuscript and your responses to the reviewers' comments. Please include in your rebuttal if you found it impossible to address certain comments. The revised version will be inspected by the editors and reviewers.

If the reviewers have suggested that your manuscript should undergo extensive English editing, please address this during revision. We suggest that you have your manuscript checked by a native English speaking colleague or use a professional English editing service. Alternatively, MDPI provides an English editing service checking grammar, spelling, punctuation and some improvement of style where necessary for an additional charge (extensive re-writing is not included), see details athttps://www.mdpi.com/authors/english.

 # One of the co-authors (T.A.N) is a native English speaker and made further improvements to the text, including the new revisions.

Do not hesitate to contact us if you have any questions regarding the revision of your manuscript or if you need more time. We look forward to hearing from you soon.

Kind regards,

Brenda Zhao

Review Report Form 1

Comments and Suggestions for Authors

I do not see, why you need two aims, at least in relation to study title. The first aim sounds more as objective.

# We considered your suggestion, but in our opinion both aims are important and both parts (understory vegetation and disturbances) are directly addressed in the title.

How you deal with the edge effect, how you distinguish it between the effects of disturbances.

# In the improved version of the ms, the evaluation of disturbance effects was substantially different than in the previous version. In the present version only direct effects of disturbances on the plot and their buffer zones were considered. Edge effects caused by disturbances were considered in plots and their buffer zones. Overall, the stands are quite closed and located within larger contiguous forest areas with minor edge effects from other land-use or forest openings.

On lines136-138 you mentioned significant deer browsing impact on forest natural regeneration, how does this has taken into account in the assessment of Stand Disturbance Index.

# The deer browsing impact, which is very significant on almost all IM sites, was not considered in this study. Deer browsing was not detected within the disturbance evaluation method used in our study. This method only identified significant disturbances that caused visible damages to some component of the stand, including mortality of entire trees, damage to tree crowns, salvage logging of damaged or dead trees, and disturbance to ground-surface layers, including the upper-soil layers and ground-vegetation layer. Deer browsing was only mentioned as one of the disturbances factors, but it was not evaluated in our study. For the evaluation of deer browsing effects, smaller plots are commonly used in which a more detailed approach is applied. At the site or plot scale studied in this study it is not possible to evaluate these effects. Indeed, it is possible that deer browsing may have influenced the understory herb layer, but the deer population has remained stable in Slovenia over the past decade, so even though browsing pressure is strong, we suspect that there was not a notable change in this driver during the study period.

Lines 239-245 does in the correlation of SDI and different structural and diversity variables has been considered site specific impacts?

# We removed the calculation of the SDI index and correlations among different variables from the improved version of the ms. A more detailed approach estimating disturbance impacts in a percentage scale was applied in the improved version.

Conclusions is missing, but what is the take home message? what are the the direction of changes.

# We added a Conclusion section.

Reviewer 2 Report

This is an interesting article with important implications for forest management and for understanding the effects of natural and human-caused disturbance on forest biodiversity. The most important contribution of the study is the measurement of forest structure and composition over a 10 year window where there was a notable increase in forest disturbance as measured by the authors’ SDI index. The manuscript is generally well written and organized and makes good use of the peer-reviewed literature on the subject. The length of the manuscript is appropriate for the content provided. Although the manuscript is presented effectively, there are several changes that need to be implemented for accuracy and to make the manuscript stronger. These are outlined here with some additional minor comments below.

The following are major issues with the manuscript that need to be addressed.

1) Pseudoreplicated data were used for most of the statistical analyses in the manuscript. This is inappropriate, as these data violate the assumption of independence among sample units for all tests and inflate the probability of type I statistical error. All of the n=60 sample plots were used in the calculation of Wilcoxon Matched Pairs Test (Table 3 and Table 4), and for the calculation of Spearman Rank Correlation (Table 5). The level of the independent sampling unit in this study is the n=10 sites. The plots within each study site are simply sub-samples (i.e., pseudoreplicates). All tests will need to be re-calculated at the site-level in order to meet the most basic and required assumption of the tests used.

2) Spearman Rank Correlations performed for Table 5 are difficult to interpret because the calculation of SDI-di does not have equal differences among values. Namely,

di=1 has less than 10% disturbance

di=2 has between 10-20% disturbance

di=3 has between 20-30% disturbance and

di=4 has between 30-100% disturbance.

Therefore, when performing correlations, different values of forest attributes do not represent linear changes in SDI-di, and this makes correlations difficult to interpret. This appears to be an inherent problem with the SDI index but there may be other ways of presenting the data (at the non pseudoreplicated site-level) that are more meaningful.

3) Although species and forest data were collected between P1 and P2, there is no indication in the manuscript about whether the plots were re-measured in the exact same location at both times. This would make a difference in the interpretation of results. Differences in plot locations could account for some of the variation in species composition between time periods.

Also, there is no indication about the individuals who recorded the data between time periods. Different recorders/observers could provide very different estimates of percent cover if not properly trained and calibrated to each other, and they may bring very different experience when it comes to plant identification.

4) Both Shannon H’ and Simpson D were used as diversity measures (P7, L178). These indices measure very different aspects of species diversity in ecosystems but there was no explanation for why both measures were chosen for the study. Without justification for these measures it appears as though the authors are analyzing factors without considering them carefully.

5) Species data are analyzed at a very basic level (frequency and cover) when they could potentially provide more detailed information about the study sites over time. I recommend performing unconstrained multivariate ordination on the species data (e.g., correspondence analysis, principal coordinates analysis) to gauge the magnitude of compositional differences among the n=10 sites and between P1 and P2. Complementary tests such as multiple response permutation procedures (MRPP) or permutational multivariate analysis of variance (PerMANOVA) can test statistically the differences in composition between species groups, such as differences between P1 and P2.

6) Wilcoxon Matched Pairs Test was used in Table 4 to detect species that differed in cover or between P1 and P2. However, indicator species analysis (ISA) is a widely used and highly-cited statistical tool to conduct these tests more appropriately (i.e., using permuation for tests of significance). ISA should be used instead of Wilcoxon Matched Pairs Test for testing individual species differences.

Further, it would be extremely informative to divide species into functional groups (e.g., closed canopy specialists, ruderal species) or rank them using Ellenberg indicator (e.g., moisture, light) values, to see which species are responding to forest disturbance in more detail.

7) The formula used to calculate SDI-di on P6 is not clear and should be explained more carefully. Why is D weighted by kD=0.25 and di weighted by kd=0.75? Would this formula would change when there are 8 plots within a site?

8) Page 7, L168-171 and L172-175. These paragraphs are not clear and should be clarified. Need to provide specific details for these measurements.

9) The Discussion section needs to be revisited to expand on details for many statements (see below).

Minor comments

Abstract L11 – “placed across different managed forests”; more explanation could be provide here as the statement is otherwise too broad.

Abstract. L22 – SDI index values are provided without any explanation about their range or meaning.

Abstract L25 – A “lagged response” of the understory is mentioned in the abstract but this is not followed up on in the Discussion. This is interesting, especially since disturbances within forests could have occurred anytime between P1 and P2. This means that different forest sites may be at different stages of recovery or development since disturbance.

Introduction, Page 1, L38. The Introduction mentions that the understory includes “a wide variety of growth forms and functional groups” but does not present or analyze these in the manuscript. This sort of classification of species would be important for characterizing the changes in species composition and diversity over the two time periods.

Page 2, L64-66. This sentence is unclear and could be rewritten for clarity.

P2, L85. What are “sanitary fellings”?

P2, L89. “Up to 3 hectares was selected”. What is the range of plot sizes that were established? These would be important to include in Table 1 as they impact the area used to determine D for SDI index.

P2, L89-90. Explain why 4 or 8 plots were chosen. Higher numbers of plots within a site could influence the numbers of species detected because of the greater area that is sampled.

Table 1, P3. For the Elevation column, include “meters” in the header and remove “m” from all of the values for sites in that column.

P4, L100. “A complete survey of vegetation” was conducted but bryophytes are not included as part of the vegetation in the report. Perhaps just include “vascular plants” as the vegetation that was sampled.

P5, L124. Explain what is meant by “degradative processes”.

Table 2, P5. Include a column with the SDI index that was calculated for each site for P1 and P2.

P7, L192. “Plant species per plot slightly increased between P1 and P2 (37.55 to 37.72 species)”. Values are reported at too many significant digits and there is no difference (instead of a slight increase!).

Table 3. Analyses here are based on pseudoreplicated data and need to be re-analyzed at the site-level. See major comment above.

Table 3. Values for cover are presented at too many significant digits (should be limited to one).

P9, L211-213. “Several species significantly increased in cover”. The possible reasons for this should be presented in the Discussion.

P9, L216-217. The possible reasons for this should be presented in the Discussion.

Table 4. Analyses here are based on pseudoreplicated data and should also be analyzed by an appropriate statistical tool such as ISA. See major comment above. A new table of species that were lost from sites or that were gained at sites would be insightful.

Table 5. Analyses here are based on pseudoreplicated data and need to be re-analyzed at the site-level. See major comment above.

Discussion, P11, L255-256. The differences between Shannon and Simpson indices should be explained and how these relate to the study findings. Both indices measure different aspects of species diversity and should be justified in the Methods section in terms of the ways they measure diversity and how they relate to study objectives.

The Discussion section should provide more specific details about the types of species that were lost and how this could influence regional biodiversity – one of the strengths of the paper.

P12, L270. Provide more details about these existing competitive species.

P12, L281. This statement is too general. Provide more details.

P12, L285. Change “week” to “weak”.

P12, L310-311. “The 10-year data presented here contrasts with this notion”. This statement is misleading should be removed. The 10-year data from this study are not comparable to the study from Central Europe. The SDI-di index includes both natural and human-caused disturbance in forests, whereas the previous study in Central Europe was looking at sustainable forest management only (and likely not the effects of natural disturbances in combination with human-caused ones).

P12, L313. “Marked increase in disturbance over the past decade”. This needs a citation.

P13, L317. “Biodiversity is predicted to benefit from intensified disturbance regimes”. This statement is counter to many previous studies and is very misleading! The increase in biodiversity is likely attributed to disturbance adapted species or early succession species following disturbance, but this will also depend on when the disturbance occurred and its severity within a forest stand. The increased biodiversity from disturbance is likely only short term because of the debt that still needs to be paid from the response of sensitive species to disturbance. None of this was analyzed for this study. By ending the paper on this unfounded statement, the authors are advocating that disturbance is beneficial for biodiversity, sending a dangerous message that is not based on fact.

Appendix. Include a table of all species with frequency by site for both P1 and P2. The table could also include other traits for species (e.g., Ellenberg indicator values) if you decide to include those (recommended).

Author Response

Manuscript ID: forests-583328

Title: Effects of disturbance on understory vegetation across Slovenian forest ecosystems

Authors: Lado Kutnar *, Thomas Andrew Nagel

Received: 13 August 2019

E-mails: lado.kutnar@gozdis.si, tom.nagel@bf.uni-lj.si Submitted to section: Forest Ecology and Management, https://www.mdpi.com/journal/forests/sections/Ecology_Management

------------------------------------------

Instruction for authors: 

Ø Authority on Latin binomial should be provided after each common name the first time referred to in the text.

# We added the authors of all Latin binomial species names addressed in our manuscript. Authority of Latin binomial names were provided in the text, and in Table 4 where the most common species are listed.

Ø The number of significant digits should be based on the precision of the analytical method and be rounded accordingly, and the variables presented should be correct and consistent.

# The methods and evaluation of variables suggested by the editors and reviewers were significantly improved. The number of digits in many cases were reduced according to the method used.

Ø Conclusions: this section is mandatory and should be added to the manuscript.

# OK, we added a Conclusion section to summarize the main focus of the study and suggestions for further, long-term studies.

------------------------------------------

Your co-authors can also view this link if they have an account in our submission system using the e-mail address in this message.

Please revise the manuscript according to the reviewers' comments and upload the revised file within 10 days. Use the version of your manuscript found at the above link for your revisions, as the editorial office may have made formatting changes to your original submission. Any revisions should be clearly highlighted, for example using the "Track Changes" function in Microsoft Word, so that changes are easily visible to the editors and reviewers. Please provide a cover letter to explain point-by-point the details of the revisions in the manuscript and your responses to the reviewers' comments. Please include in your rebuttal if you found it impossible to address certain comments. The revised version will be inspected by the editors and reviewers.

If the reviewers have suggested that your manuscript should undergo extensive English editing, please address this during revision. We suggest that you have your manuscript checked by a native English speaking colleague or use a professional English editing service. Alternatively, MDPI provides an English editing service checking grammar, spelling, punctuation and some improvement of style where necessary for an additional charge (extensive re-writing is not included), see details athttps://www.mdpi.com/authors/english.

 # One of the co-authors (T.A.N) is a native English speaker and made further improvements to the text, including the new revisions.

Do not hesitate to contact us if you have any questions regarding the revision of your manuscript or if you need more time. We look forward to hearing from you soon.

Kind regards,

Brenda Zhao

Review Report Form 2

Comments and Suggestions for Authors

This is an interesting article with important implications for forest management and for understanding the effects of natural and human-caused disturbance on forest biodiversity. The most important contribution of the study is the measurement of forest structure and composition over a 10 year window where there was a notable increase in forest disturbance as measured by the authors’ SDI index. The manuscript is generally well written and organized and makes good use of the peer-reviewed literature on the subject. The length of the manuscript is appropriate for the content provided. Although the manuscript is presented effectively, there are several changes that need to be implemented for accuracy and to make the manuscript stronger. These are outlined here with some additional minor comments below.

# All parts have been thoroughly revised and significantly reformulated.

The following are major issues with the manuscript that need to be addressed.

1) Pseudoreplicated data were used for most of the statistical analyses in the manuscript. This is inappropriate, as these data violate the assumption of independence among sample units for all tests and inflate the probability of type I statistical error. All of the n=60 sample plots were used in the calculation of Wilcoxon Matched Pairs Test (Table 3 and Table 4), and for the calculation of Spearman Rank Correlation (Table 5). The level of the independent sampling unit in this study is the n=10 sites. The plots within each study site are simply sub-samples (i.e., pseudoreplicates). All tests will need to be re-calculated at the site-level in order to meet the most basic and required assumption of the tests used.

 # Indeed, this was a major oversight. The analyses have been redone on the site level to remove pseudoreplication. All the main tests and assumptions/conclusions are based on the site-level (n=10 sites).

2) Spearman Rank Correlations performed for Table 5 are difficult to interpret because the calculation of SDI-di does not have equal differences among values. Namely,

di=1 has less than 10% disturbance

di=2 has between 10-20% disturbance

di=3 has between 20-30% disturbance and

di=4 has between 30-100% disturbance.

Therefore, when performing correlations, different values of forest attributes do not represent linear changes in SDI-di, and this makes correlations difficult to interpret. This appears to be an inherent problem with the SDI index but there may be other ways of presenting the data (at the non pseudoreplicated site-level) that are more meaningful.

# This part has been completely revised. Calculation of the SDI index has been removed. The proportion of forest stand and area of ground-surface layer significantly damaged from different disturbance factors was visually estimated in a percentage scale (%), ranging from 0 to 100%. At the forest stand level, the significant disturbance impacts were estimated at each plot and its buffer zone. The buffer zone of the plot was situated around the periphery of the studied area, and the distance from the plot margin to outer margin of a buffer zone was 5 meters. All calculations are based on evaluation of disturbances impacts expressed in percentage.

3) Although species and forest data were collected between P1 and P2, there is no indication in the manuscript about whether the plots were re-measured in the exact same location at both times. This would make a difference in the interpretation of results. Differences in plot locations could account for some of the variation in species composition between time periods.

Also, there is no indication about the individuals who recorded the data between time periods. Different recorders/observers could provide very different estimates of percent cover if not properly trained and calibrated to each other, and they may bring very different experience when it comes to plant identification.

# Thanks for pointing this out. We added this clarification. The position of the 60 plots did not change between the two monitoring periods (P1, P2), and the exact the same area was re-surveyed in the second monitoring period. We also added an explanation regarding observers. In both monitoring periods, all identifications of plant species, estimations of plant cover and disturbances impacts, and all other records were done by the same observer. The observer was first author of paper (with some field assistance of his colleagues), who has almost 30 years of experiences in field of phytosociology and botany.

4) Both Shannon H’ and Simpson D were used as diversity measures (P7, L178). These indices measure very different aspects of species diversity in ecosystems but there was no explanation for why both measures were chosen for the study. Without justification for these measures it appears as though the authors are analyzing factors without considering them carefully.

# Yes, this is a good point. In the present version of the ms only the Shannon diversity index was calculated and presented.

5) Species data are analyzed at a very basic level (frequency and cover) when they could potentially provide more detailed information about the study sites over time. I recommend performing unconstrained multivariate ordination on the species data (e.g., correspondence analysis, principal coordinates analysis) to gauge the magnitude of compositional differences among the n=10 sites and between P1 and P2. Complementary tests such as multiple response permutation procedures (MRPP) or permutational multivariate analysis of variance (PerMANOVA) can test statistically the differences in composition between species groups, such as differences between P1 and P2.

# We implemented this suggestion. For the entire plot-by-species matrix, we performed an unconstrained ordination Principal Coordinates Analysis (PCoA). To test the effect of disturbance on temporal change in vascular plant composition, a simple linear regression was performed based on the averaged site-level data.

6) Wilcoxon Matched Pairs Test was used in Table 4 to detect species that differed in cover or between P1 and P2. However, indicator species analysis (ISA) is a widely used and highly-cited statistical tool to conduct these tests more appropriately (i.e., using permuation for tests of significance). ISA should be used instead of Wilcoxon Matched Pairs Test for testing individual species differences.

# We also considered this suggestion. Even before the submission of the first version, the indicator species analysis (ISA) had already been done. But it seems that due to very small changes in species composition, the ISA method did not reveal any notable patterns that we were able to interpret. Therefore, we decided to not include ISA in the previous version of ms.

We repeated the ISA once more (in R) and the results are shown below, but again we were not able to provide a meaningful interpretation. Therefore, we decided to keep Table 4 in the revised version that shows the frequency and cover of common plant species in two monitoring periods, and Fig. 3 that shows mean cover of the most common understory plant species.

Further, it would be extremely informative to divide species into functional groups (e.g., closed canopy specialists, ruderal species) or rank them using Ellenberg indicator (e.g., moisture, light) values, to see which species are responding to forest disturbance in more detail.

# We considered this suggestion too. We checked possibilities to study the Ellenberg indicator values and functional traits, but due to slight changes in species composition in the 10-year period, the changes of these variables were very small and not significant.

Analysis of functional groups revealed some changes in composition (see Graph 1 and 2 below), but they were not very informative. Thus, we decided to not add these analyses.

Graph 1: Cover (%) of functional types for period 1 (P1: 2004/2005) and period 2 (P2: 2014/2015). The plot-level percentage cover of each functional type was calculated by summing the abundances of all species belonging to a particular functional type. Wilcoxon rank sum test was used to test statistical differences between P1 and P2, using averaged site-level values (N = 10). Error bars represent standard error. Asterisk denotes significant difference (p < 0.05); ns – not significant.

Graph 2: Mean number of plant species for each functional group, separately for P1 and P2. Based on site-level values (N = 10), Wilcoxon rank sum test was used to check whether the differences between P1 and P2 are statistically significant. Error bars represent standard error. Asterisk denotes significant differences (p < 0.05); ns – not significant.

7) The formula used to calculate SDI-di on P6 is not clear and should be explained more carefully. Why is D weighted by kD=0.25 and di weighted by kd=0.75? Would this formula would change when there are 8 plots within a site?

# This part was reformulated accordingly. The SDI-di index was not used in the improved version. All calculations are based on evaluation of disturbances impacts expressed in percent.

8) Page 7, L168-171 and L172-175. These paragraphs are not clear and should be clarified. Need to provide specific details for these measurements.

# This section was reformulated, and due to a completely different approach (without SDI-di index) it was mostly deleted.

9) The Discussion section needs to be revisited to expand on details for many statements (see below).

# OK, The Discussion section has been reworked.

Minor comments

Abstract L11 – “placed across different managed forests”; more explanation could be provide here as the statement is otherwise too broad.

# We added more explanation to clarify this.

Abstract. L22 – SDI index values are provided without any explanation about their range or meaning.

# SDI-di index was not used in the improved version of ms.

Abstract L25 – A “lagged response” of the understory is mentioned in the abstract but this is not followed up on in the Discussion. This is interesting, especially since disturbances within forests could have occurred anytime between P1 and P2. This means that different forest sites may be at different stages of recovery or development since disturbance.

# This aspect is now briefly addressed in the Discussion.

Introduction, Page 1, L38. The Introduction mentions that the understory includes “a wide variety of growth forms and functional groups” but does not present or analyze these in the manuscript. This sort of classification of species would be important for characterizing the changes in species composition and diversity over the two time periods.

# OK, we considered your suggestion and tested functional groups but these results were not really compelling and we decided to not include it in the revised version (see Graph 1 and 2 above).

Page 2, L64-66. This sentence is unclear and could be rewritten for clarity.

# This was rewritten to be clearer.

P2, L85. What are “sanitary fellings”?

# This term was replaced by the term ‘salvage logging’ which is more widely understood in the literature.

P2, L89. “Up to 3 hectares was selected”. What is the range of plot sizes that were established? These would be important to include in Table 1 as they impact the area used to determine D for SDI index.

# We considered your opinion. In the new version of the ms, D (disturbances at site level) was not included in the analysis. We were focused on direct disturbances at the plot and buffer zone. This question is not relevant anymore.

P2, L89-90. Explain why 4 or 8 plots were chosen. Higher numbers of plots within a site could influence the numbers of species detected because of the greater area that is sampled.

# OK, we agree. Short explanation of this was already given in the following sentence: “The number of plots per site depends on the intensity and type of monitoring activities.” Different number is associated with some other monitoring activities which were proposed by ICP Forests programme. Higher number of plots were established at more intensive sites which are more equipped for monitoring of different variables, e.g. deposits, soil solution, dendrometers.

Table 1, P3. For the Elevation column, include “meters” in the header and remove “m” from all of the values for sites in that column.

# This was corrected.

P4, L100. “A complete survey of vegetation” was conducted but bryophytes are not included as part of the vegetation in the report. Perhaps just include “vascular plants” as the vegetation that was sampled.

# Corrected.

P5, L124. Explain what is meant by “degradative processes”.

# We provided an explanation.

Table 2, P5. Include a column with the SDI index that was calculated for each site for P1 and P2.

# SDI index is not calculated and presented in the improved version. Estimations of disturbances impacts are one of the results of this study, and they were calculated at the site-level for both monitoring periods and were shown in the chapter 3.2 Disturbances impacts.

P7, L192. “Plant species per plot slightly increased between P1 and P2 (37.55 to 37.72 species)”. Values are reported at too many significant digits and there is no difference (instead of a slight increase!).

# OK, it was considered

Table 3. Analyses here are based on pseudoreplicated data and need to be re-analyzed at the site-level. See major comment above.

# OK, we considered your suggestion and re-analysed all variables at the site-level. Changes at the site-level and plot-level are rather comparable, except the significant level is slightly lower for the most studied variables in case of sites.

Table 3. Values for cover are presented at too many significant digits (should be limited to one).

# OK, we considered your comment.

P9, L211-213. “Several species significantly increased in cover”. The possible reasons for this should be presented in the Discussion.

# We added an additional explanation in the Discussion.

P9, L216-217. The possible reasons for this should be presented in the Discussion.

# It was addressed in the Discussion as a homogenization process.

Table 4. Analyses here are based on pseudoreplicated data and should also be analyzed by an appropriate statistical tool such as ISA. See major comment above. A new table of species that were lost from sites or that were gained at sites would be insightful.

# We did ISA analysis. However, the results were not very indicative. Only few species (3) were significant in this analysis (see table in yellow above). It seems that due to very small changes in species composition, the ISA method did not determine an easily interpretable pattern. Therefore, we decided to keep Table 4 that indicating the frequency and cover of common plant species in the two monitoring periods, and Fig. 4 that indicating mean cover of the most common understory plant species in improved version.

Table 5. Analyses here are based on pseudoreplicated data and need to be re-analyzed at the site-level. See major comment above.

# OK, we considered your suggestion. The data have been analysed at the site-level (n=10).

Discussion, P11, L255-256. The differences between Shannon and Simpson indices should be explained and how these relate to the study findings. Both indices measure different aspects of species diversity and should be justified in the Methods section in terms of the ways they measure diversity and how they relate to study objectives.

# In the present version, only Shannon index was calculated and presented.

The Discussion section should provide more specific details about the types of species that were lost and how this could influence regional biodiversity – one of the strengths of the paper.

# This part was additionally analysed. However, we did not find any clear pattern that could give us additional explanation of changes.

P12, L270. Provide more details about these existing competitive species.

# Additional explanation was provided.

P12, L281. This statement is too general. Provide more details.

# This statement is explained more into detail in the following sentences.

P12, L285. Change “week” to “weak”.

#OK, it was changed.

P12, L310-311. “The 10-year data presented here contrasts with this notion”. This statement is misleading should be removed. The 10-year data from this study are not comparable to the study from Central Europe. The SDI-di index includes both natural and human-caused disturbance in forests, whereas the previous study in Central Europe was looking at sustainable forest management only (and likely not the effects of natural disturbances in combination with human-caused ones).

# This statement was removed, and additional explanation was provided to be clearer.

P12, L313. “Marked increase in disturbance over the past decade”. This needs a citation.

# We added the citation of relevant references.

P13, L317. “Biodiversity is predicted to benefit from intensified disturbance regimes”. This statement is counter to many previous studies and is very misleading! The increase in biodiversity is likely attributed to disturbance adapted species or early succession species following disturbance, but this will also depend on when the disturbance occurred and its severity within a forest stand. The increased biodiversity from disturbance is likely only short term because of the debt that still needs to be paid from the response of sensitive species to disturbance. None of this was analyzed for this study. By ending the paper on this unfounded statement, the authors are advocating that disturbance is beneficial for biodiversity, sending a dangerous message that is not based on fact.

# Thank you pointing this out. We should have been more careful in this interpretation. We considered your suggestion and clarified this statement.

Appendix. Include a table of all species with frequency by site for both P1 and P2. The table could also include other traits for species (e.g., Ellenberg indicator values) if you decide to include those (recommended).

# The list of all species in this study is quite high (>250), and some of them have very low frequency and cover that might be less interesting for the international audience. In the Table 1 and in the description of study area, the main tree species are listed. Beside that we decide to keep the table with list of more than 30 most common species that occur on 25% or more of the plots in one period at least. We believe that these might be the most informative and interesting species.

We also checked possibilities to study the Ellenberg indicator values, but due to slight changes in species composition in 10-years period, the changes in Ellenberg values are rather small and not significant.
